# Photo-induced enhanced Raman spectroscopy for universal ultra-trace detection of explosives, pollutants and biomolecules

Sultan Ben-Jaber[1], William J. Peveler[1], Raul Quesada-Cabrera[1], Emiliano Cortés[2], Carlos Sotelo-Vazquez[1], Nadia Abdul-Karim[1], Stefan A. Maier[2] & Ivan P. Parkin[1]

Surface-enhanced Raman spectroscopy is one of the most sensitive spectroscopic techniques available, with single-molecule detection possible on a range of noble-metal substrates. It is widely used to detect molecules that have a strong Raman response at very low concentrations. Here we present photo-induced-enhanced Raman spectroscopy, where the combination of plasmonic nanoparticles with a photo-activated substrate gives rise to large signal enhancement (an order of magnitude) for a wide range of small molecules, even those with a typically low Raman cross-section. We show that the induced chemical enhancement is due to increased electron density at the noble-metal nanoparticles, and demonstrate the universality of this system with explosives, biomolecules and organic dyes, at trace levels. Our substrates are also easy to fabricate, self-cleaning and reusable.

[1] Department of Chemistry, University College London, 20 Gordon St, London WC1H 0AJ, UK. [2] The Blackett Laboratory, Department of Physics, Imperial College, London SW7 2AZ, UK. Correspondence and requests for materials should be addressed to I.P.P. (email: i.p.parkin@ucl.ac.uk).

Surface-enhanced Raman spectroscopy (SERS) is one of the most popular and powerful techniques in analytical chemistry, with single-molecule detection potentially accessible with noble-metal structures or substrates[1–5]. The adsorption of analyte molecules to a roughened metal surface (typically a gold or silver nanoparticle) leads to strong enhancement of the analyte's Raman signals. The low limits of detection enable trace analysis of highly important molecules such as explosives, pesticides and biomarkers for the protection and preservation of human health[6–8].

The two key contributions to the SERS enhancement are the electromagnetic (EM) factor and the chemical contribution[9]. The EM enhancement of SERS-active substrates is by far the most important, and it is mainly determined by the nanostructure of the metallic surface and the wavelength-dependent dielectric properties of the metal. The conduction electrons of these nanoscale features can be driven by the incident electric field in collective oscillations known as localized surface plasmon resonances (LSPRs). These plasmonic materials are able to localize the electromagnetic field at sub-wavelength scales, thus effectively acting as nano-antennas[10–13]. However, the difficulty in arranging nanoparticles on a substrate to create the ideal SERS hotspots, means that the possible sensitivities are often not obtained in an easy or repeatable fashion[14]. On the other hand, the chemical contribution has been associated with smaller additional enhancements and its microscopic origin and overall contribution to the SERS enhancement are still under debate. Several models have been put forward depending on the specific system under study, including the adatom[15], the metal-molecule charge transfer resonance[9,16,17] and the polarizability modulation models[18,19]. These models were proposed to explain enhancements in different systems, but there are few demonstrations on how to induce and generalize this mechanism/effect.

SERS on (non photo-activated) semiconductor materials has been explored[20–22]. It has been shown that the main enhancement mechanism is non-plasmon based[23,24], with enhancement factors, until recently, lower than those reported for metallic substrates[25]. The incorporation of metallic particles onto these (non-active) substrates leads to hot-electron migration from the particles to the semiconductor under visible light illumination[26], with no effect on the SERS signal beyond that arising from the metallic particles alone[27]. There has also been limited work on the incorporation of semiconductor nanoparticles on roughened gold SERS substrates for enhancement[28].

Meanwhile, catalytic applications for ultraviolet light photo-activated $TiO_2$ have been extensively explored[29], including the incorporation of metallic nanoparticles[30]. Core–shell charge transfer between Ag cores with a $TiO_2$ shell has been demonstrated by Kamat et al. with ultraviolet irradiation[31]. However, covering the metallic core limits the applications for binding analytes directly to the silver particles. This has detrimental consequences for the SERS enhancement factors.

For plasmonic applications—such as SERS—the molecules must be as close as possible to the metal surface as EM enhancement decays exponentially with the distance ($d$) from surface[2]. The chemical contribution implies charge transfer between the substrate and the molecule, thus also depends on $d$.

We propose here photo-induced SERS enhancement, whereby pre-irradiation (photo-excitation) of a semiconductor, such as $TiO_2$ enables strong Raman enhancement at the nanoparticle sites, increasing sensitivity beyond the normal SERS effect. We call this effect photo induced enhanced Raman spectroscopy—PIERS. In our system the molecules can be directly adsorbed to the metallic particle, allowing controlled enhancement by both the EM factor and chemical enhancement factor. As there is no requirement for simultaneous ultraviolet irradiation, which may cause analyte degradation, and our system is applicable to a wide range of engineered gold SERS substrates, thus the PIERS effect may become a useful method of enhancing Raman spectra.

## Results

**Substrate synthesis.** To create our PIERS substrates, citrate-capped gold nanoparticles (AuNPs) (27 nm, s.d. of 5 nm, LSPR of 520 nm in water) or silver nanoparticles (AgNPs) (58 nm, s.d. of 14 nm, LSPR of 428 nm in water) were deposited from MeOH/$H_2O$ on a $TiO_2$ rutile (R) surface formed by atmospheric pressure chemical vapour deposition (APCVD). The thickness of the $TiO_2$ was ca. 500 nm, and the final particle concentration was ca. 250 particles per $\mu m^2$ (Fig. 1). This substrate was then irradiated with ultraviolet light (254 nm—UVC) for a period of time, followed by deposition of the analyte sample and Raman analysis using a 633 nm excitation laser (1.9 mW). We chose both Au and Ag particles for our analysis, to disentangle electromagnetic and chemical enhancement effects, as on $TiO_2$ only the Au LSPR resonance will be in better resonance with the 633 excitation laser line. A schematic is presented in Fig. 2. In the first instance Rhodamine 6G (R6G) was used as a standard (and commonly applied) analyte and significant enhancements over spectral intensities on non-irradiated substrates (those demonstrating normal SERS) were seen (Fig. 3a). Physical characterization of the $TiO_2$ substrate was performed before and after irradiation, and no change in composition was observed by X-ray diffraction (Supplementary Fig. 1). X-ray photoelectron spectroscopy confirmed the presence of the gold particles. (Supplementary Fig. 2).

**PIERS effect and mechanism.** The mechanism of action for PIERS is proposed to be ultraviolet light mediating electron migration from the semiconductor substrate, $TiO_2$ (R), to the metallic nanoparticle (Fig. 2): the PIERS effect is based on carrier migration and separation from the active $TiO_2$ (R) surface to the AuNPs. It has been demonstrated that pre-irradiation with UVC wavelengths can create oxygen vacancy ($V_O$) defects on $TiO_2$ surfaces[32]. These $V_O$ species produce donor states at ~0.7 eV below the $TiO_2$ conduction band edge[33] and induce optical

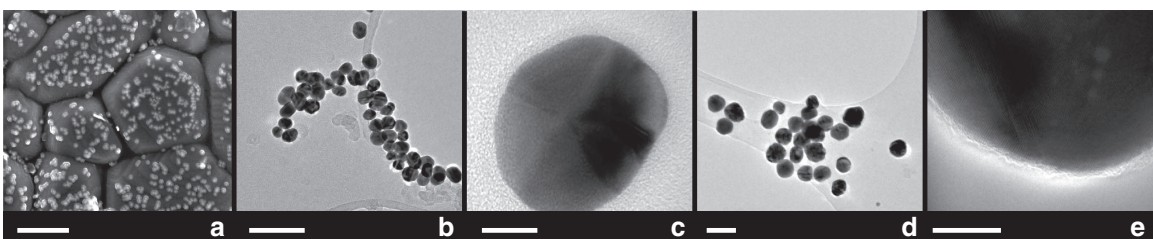

**Figure 1 | Electron microscopy of substrate and particles.** (**a**) AuNPs on $TiO_2$ (**b,c**) AuNPs, (**d,e**) AgNPs. Scale bars given are 500 nm (**a**), 100 nm (**b,d**) and 10 nm (**c,e**).

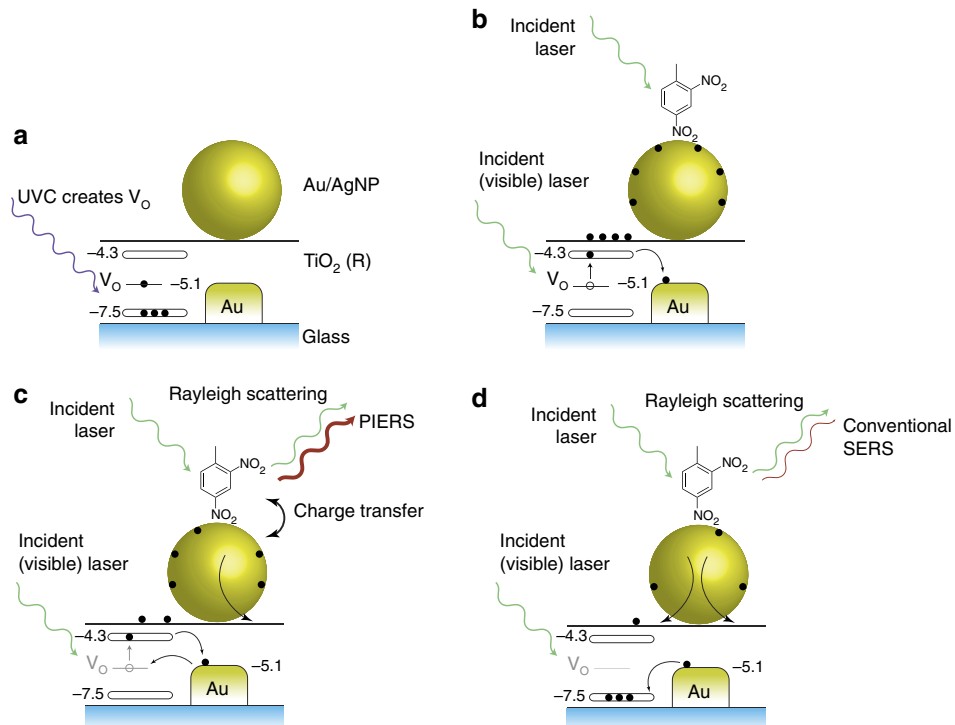

**Figure 2 | Photo-induced enhanced Raman spectroscopy (PIERS).** (**a**) Ultraviolet C light (UVC) pre-irradiation creates oxygen vaccencies ($V_O$) in the $TiO_2$ below the conduction band. (**b**) The sample (here dinitrotoluene) is deposited and irradiation with the Raman laser (633 nm) photoexcites the $TiO_2$, leading to increase in charge on the nanoparicles. (**c**) The charged nanoparticless lead to the PIERS enhancement. As the $V_O$ are replenished, upon exposure to air, fewer electrons can be photoexcited into the conduction band of $TiO_2$. (**d**) The PIERS effect gradually disappears as the $V_O$ are completely replenished over time (30–60 min). The substrate can then be cleaned and recharged through exposure to more UVC light.

absorption above 500 nm (ref. 34). In the PIERS effect, electrons will thus be promoted from the $V_O$ states into the conduction band of $TiO_2$ upon Raman laser illumination and then transferred into Au energy levels[35]. The process will slowly decay, with a lifetime of approximately an hour, due to surface healing upon exposure of the substrate to air[36]. Indeed, our results show an intensity decrease of the PIERS signal occurring within the initial 30–60 min (Fig. 3c). This long lifetime of the excited carriers on the Au nanoparticles allows us to evaluate their influence on SERS/PIERS after substrate activation with UVC pre-irradiation.

To demonstrate the activation of the AuNPs we measured the absorption spectra of the substrates before and after UVC irradiation. As shown in Fig. 3b, a 12 nm blue-shift of the AuNP LSPR is observed on irradiation, due to the increasing electron density on the particles[31]. In conjunction to this, the full width at half-maximum (FWHM) of the LSPR increases from 72 to 88 nm on irradiation. This is in agreement with the work of Brown *et al.*[37] who, in bias-dependent experiments of Au nanoparticles deposited on indium-tin oxide substrates, also observed a blueshift, increased intensity and broadening of the FWHM of the LSPR peak when a negative potential is applied, confirming our idea of a light-induced charge transfer process from $TiO_2$ to the AuNPs. Following the approach of Mulvaney *et al.*[38] we can estimate the injected electron density ($\Delta N/N$) on the AuNPs due to $TiO_2$ pre-irradiation as follows:

$$\frac{\Delta N}{N} = -\frac{2\Delta\lambda}{\lambda_0} \quad (1)$$

Where $\Delta\lambda$ is the measured wavelength shift ($\sim -12$ nm) and $\lambda_0$ is the initial AuNP plasmon peak position on the $TiO_2$ substrate ($\sim 605$ nm). From Fig. 3b we can extract then ($\Delta N/N$) $\approx 4\%$ (this data was taken 20 min after pre-irradiation ended).

To further demonstrate the effect, the reduction of PIERS intensity after standing the substrate in the dark was monitored over an hour, in comparison to a non-irradiated SERS film, to show how recombination of the $TiO_2$ excitons reduced the enhancement (Fig. 3c). From Fig. 3b and c, we can estimate the initial injected charge density (at time 0) is about 10%. Finally, a similar substrate was prepared using a more photo-inactive $SiO_2$ film with similar AuNP and analyte loadings, and no significant PIERS enhancement was observed after the ultraviolet pretreatment (Supplementary Fig. 5); nor was any enhancement observed in the absence of metallic nanoparticles (Supplementary Fig. 6).

The experiments presented so far demonstrate that there is a net charge transfer process to the AuNPs resulting from the activation of the $TiO_2$ substrate via pre-irradiation. We have quantified the density of transferred charge and also showed the lifetime decay by following the PIERS signal. The fact that similar PIERS enhancements are seen with AgNPs (LSPR on $TiO_2$ at 460 nm) demonstrates that the observed enhancement in signal upon ultraviolet treatment is indeed chemical in nature, in addition to a contribution from electromagnetic field enhancement effects (Supplementary Figs 7 and 8).

The (average) order of magnitude enhancement found for PIERS respect to SERS and the inhomogeneous enhancements between different peaks in the Raman spectra when comparing both, points towards an improved chemical enhancement in PIERS, beyond the typical EM effect[39]. Indeed, in both AuNP and AgNP cases, the injected charges will shift the Fermi level of the nanoparticle to more negative potentials. The exact value of the new Fermi level is dependent on the size of the nanoparticle[40], and as a consequence we expect a broad distribution of Fermi level values depending on both the intrinsic size of the nanoparticle and the amount of charge injected. This fact is also reflected in the spectral broadening of the

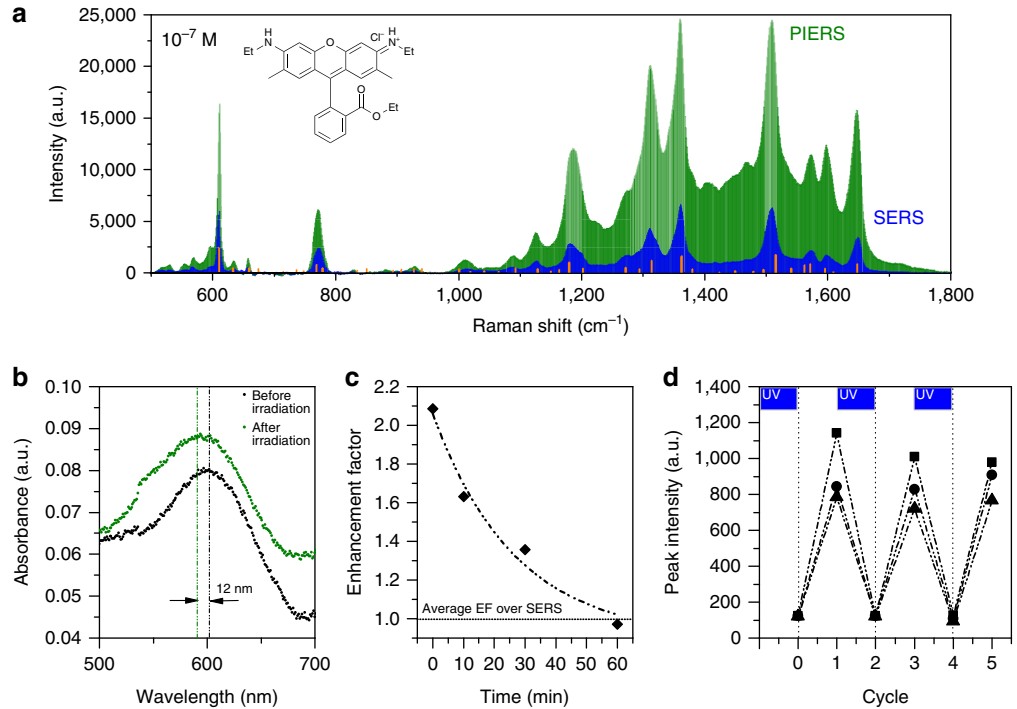

**Figure 3 | PIERS spectra and examining the PIERS mechanism.** (**a**) PIERS (green) and SERS (blue) spectra of Rhodamine 6G ($10^{-7}$ M), with Raman modes of the solid shown by the orange lines. All spectra collected under identical collection conditions. (**b**) Absorption spectra of a thin film showing LSPR shifts for AuNPs on $TiO_2$ (R) after irradiation with 254 nm light (3 h). Note that the AuNP pre-irradiation LSPR is red-shifted from its solution value, due to the high-refractive index of the $TiO_2$ surface compared with water. Dashed lines indicate peak position (**c**) Average of total spectral enhancement (EF) of PIERS over SERS for dinitrotoluene ($10^{-9}$ M) over time (full spectra in Supplementary Fig. 3). Horizontal line indicates the normal SERS intensity. (**d**) Demonstrating the recyclability of the substrate on irradiation to clean and re-charge for measuring DNT ($10^{-9}$ M) at three wavelengths, 1,315 $cm^{-1}$ (squares), 1,367 $cm^{-1}$ (circles) and 1,385 $cm^{-1}$ (triangles) (full spectra in Supplementary Fig. 4). Sample irradiation cycles with 254 nm light are marked with blue boxes, then spectra are taken before (on the vertical lines) and after adding DNT, before re-irradiating.

post-irradiated sample in Fig. 3b. Having isolated particles on the substrate avoids equilibration of the charges (Fig. 1), resulting in different individual contributions to the observed chemical enhancement in the PIERS signal, depending on each particular Fermi level shift.

To account for the influence of a broad and new AuNP Fermi level distribution in the PIERS signal (compared with the SERS signal), it is necessary to analyse the parameters involved in the molecule-to-substrate (or vice-versa) charge transfer process, (the origin of the chemical enhancement)[16]. In this way, the differential Raman cross-section in SERS can account for the microscopic contributions to the chemical enhancement mechanism. By computing this parameter, Tognalli *et al.*[9] clearly show how depositing a single Ag layer over an active Au SERS substrate results in the appearance of a new molecule Fermi level charge transfer resonance, thus enhancing the Raman signal of 4-mercaptopyridine molecules. In our case, the injected charges also shift the Fermi level of the AuNP over a broad distribution of more negative values. As a consequence, this will broaden the resonance conditions between the Fermi level and the molecular orbitals, increasing the charge transfer transitions probabilities over a wider variety of molecules deposited on the substrates.

**Detecting explosives and biomolecules.** The possibility of photo-inducing chemical enhancement of SERS, independently of the nature of the molecule, even for low Raman cross-section species, leads to a wide range of applications for this PIERS technique. An area where SERS substrates are of great value is in homeland security, for detection of high explosives during environmental

monitoring and post-blast forensics[41]. This requires SERS techniques that work on inexpensive, reusable substrates with high sensitivities for low-cross-section molecules such as nitro-aliphatic PETN (pentaerythritol tetranitrate) and RDX (cyclotrimethylenetrinitramine), as well as the widely used TNT (trinitrotoluene) and its decomposition product DNT (dinitrotoluene). Other areas of interest are small biomolecule sensing, for example in glucose monitoring, or pollution monitoring and control, with similar substrate requirements.

Initial experiments on R6G and DNT showed that the enhancement factor varied between the different bands, with some bands showing an enhancement factor of 20× over conventional SERS, and some bands were visible only in the PIERS spectra (Fig. 4; Supplementary Table 1; Supplementary Note 1). This kind of selective mode enhancement, along with small shifts in peak position, is typical of chemical enhance-ment[15,18]. Due to this PIERS enhancement, it was possible to get an excellent Raman spectrum of DNT at concentrations as low as $10^{-15}$ M (Supplementary Fig. 9).

To demonstrate the versatility of the PIERS beyond DNT and R6G, other materials of interest with low Raman cross-sections, such as pentaerythritol tetranitrate (PETN), RDX (cyclotrimethy-lenetrinitramine) (explosives) and glucose (biomarker) were tested (Fig. 4; Supplementary Fig. 10). There was significant enhancement over their solid and SERS spectra, demonstrating the power of the PIERS technique to enhance spectra of many classes of explosive molecules, not just high cross-section organic dyes or nitroaromatics.

Vapour phase detection of high explosives was attempted with a solid sample of TNT, placed 5 cm from the pre-irradiated

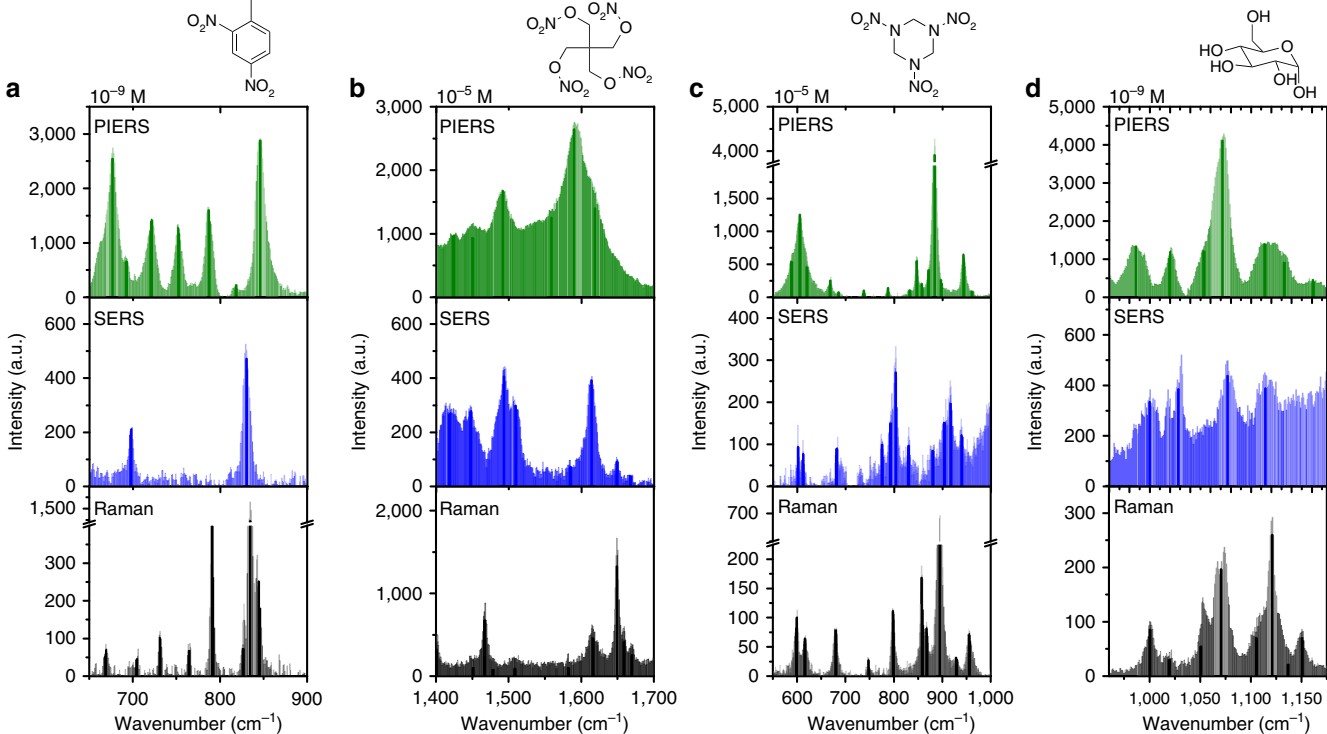

**Figure 4 | Comparative PIERS spectra of explosives and biomolecules.** Raman of pure solid (black), SERS (blue) and PIERS (green) of molecules of interest on a gold nanoparticle/TiO₂ substrate. Darker lines indicate the positions of spectral maxima to aid comparison. SERS and PIERS samples deposited from identical volume of solution: (**a**) $10^{-9}$ M dinitrotoluene (DNT), (**b**) $10^{-5}$ M pentaerythritol tetranitrate (PETN), (**c**) $10^{-5}$ M cyclotrimethylenetrinitramine (RDX), (**d**) $10^{-9}$ M glucose. Raman spectra collected from pure reference powders/liquids. Each spectrum was collected with the same conditions, giving the increase in intensity for the PIERS spectra, and full collection details are given in the Methods, along with full spectra in Supplementary Fig. 10.

substrate and the system was allowed to equilibrate for 5 min. The PIERS spectrum was then measured (Supplementary Fig. 11) showing clear TNT signals. Based on a reported vapour pressure of TNT of $7.3 \times 10^{-4}$ Pa at room temperature, a saturated atmosphere approximates to 7 p.p.b. (0.007 mg l$^{-1}$)[42]. Thus, this can be approximated to correspond to a molarity of around $3.1 \times 10^{-8}$ M (Supplementary Note 2).

Finally, the photocatalytic properties of the TiO₂ (R) were put to further use to add self-cleaning properties to the substrate. If a used substrate was further irradiated with 254 nm light then it could be cleaned of all organic residues, with the action of the hard ultraviolet and any generated surface electrons, returning it to a pristine state for further experiments. This process could be cycled without loss of analyte signal intensity (Fig. 3d; Supplementary Fig. 4).

## Discussion

We have demonstrated a powerful extension of SERS—PIERS—for the detection of ultra-trace levels of various small molecule analytes. Most important is the order of magnitude universal enhancement of the analyte signals, the recyclability of the substrates and the potential single-molecule detection that can be achieved. By combining the traditional metallic nanoparticles used for SERS, with a photoactive (pre-irradiated) TiO₂ film, enhancement factors of an order of magnitude can be achieved over traditional SERS spectra, even for low Raman cross-section analytes. We propose that the mechanism of operation for this enhancement is due to Fermi level modification of the metallic NPs on the surface, improving chemical enhancement of the SERS signal and show that this is an early example of broad applicability of the chemical enhancement. We hope that this

technique might be extended further through applying the same idea to carefully engineered plasmonic substrates. By coupling the increasing prevalence of portable Raman spectrometers and improved spectral analysis techniques with the sensitivity, robustness and inexpensive nature of PIERS, highly sensitive Raman analysis can become an integral part of not just homeland security, but also biological analysis, and environmental protection, where low Raman cross-sections can often prevent or hinder spectral monitoring.

## Methods

Films were produced by drop casting Au or AgNPs onto metal oxide thin films. To produce the AuNP/TiO₂ films detailed, 100 μl of a AuNP solution ($\sim 4 \times 10^{-10}$ M) in MeOH was dropped onto a 1 cm × 1 cm metal oxide substrate and allowed to dry in air. This gave an approximate coverage of 250 AuNPs per μm².

Films were irradiated under 254 nm light (2 × 8 W bulbs at a distance of 13 cm) for 4 h, before application of fixed volumes of methanolic solutions of explosive, created from analytical standards, methanolic solutions of R6G, or aqueous solutions of glucose. In each case a fixed drop volume of 50 μl of analyte was applied and allowed to air dry on the film. Typical SERS spectra were collected by applying the drop to the film without the pre-irradiation step.

Raman spectra were measured after air-drying the films, on a Renishaw Raman inVia microscope with a 633 nm He–Ne excitation laser (1.9 mW when operated at 25% power, spot size $\sim 4.4$ μm²). All samples were measured under identical conditions and multiple points were measured on each sample to ensure consistency between locations. Where appropriate, spectra are composites of multiple locations, however as this is a particle-enhanced technique, there were definite hotspots where stronger spectra were collected than on other regions, where weak, or no signal was observed. For direct comparison between PIERS and SERS spectra, normalisation of the Raman spectra was carried out using the standard deviations of the baseline signal for corresponding powder, SERS and PIERS spectra, within the range of 1,800–2,000 cm$^{-1}$. Data was then scaled to obtain comparable signal-to-noise ratios. When plotting all spectra within a series are scaled to the same degree.

Additional protocols for chemical vapour deposition of films and particle synthesis, along with a list of characterization methods and set up details, are detailed in the Supplementary Methods.

**Data availability.** The data that support the findings of this study are available from the corresponding author on request.

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

## Acknowledgements

We would like to thank Dr Steven Firth for useful discussion on the Raman spectra. SBJ acknowledges the support of the government of Saudi Arabia, Ministry of Interior, King Fahd Security College (KFSC), WJP is supported by EPSRC Doctoral Prize Fellowship EP/M506448/1, EC is supported by a Marie Curie fellowship. SAM and EC acknowledge the EPSRC Reactive Plasmonics project EP/M013812/1, the Office of Naval Research, the Royal Society, and the Lee-Lucas Chair in Physics.

## Author contributions

Experiments were designed by S.B-J., W.J.P., R.Q-C., E.C. and I.P.P. Materials synthesis was performed by W.J.P., C.S-V., E.C. and S.B-J. Raman experiments were performed by S.B-J. and R.Q-C., materials characterization by C.S-V., W.J.P. and R.Q-C., and plasmonic measurements by E.C. Explosives samples were provided, prepared and handled by N.A-K. Data analysis and interpretation was performed by S.B-J., W.J.P., R.Q-C., E.C., S.A.M. and I.P.P. The manuscript was written by W.J.P., R.Q-C., E.C., S.A.M. and I.P.P., with input from all the authors.

## Additional information

**Competing financial interests:** The authors declare no competing financial interests.

