## [Peer review file · Nature Communications]

REVIEWERS' COMMENTS:

Reviewer #1 (Remarks to the Author):

The authors have reorganized the manuscript, and most of the technical issues are addressed with supplementary experiments. As a rebuttal by the authors, "pre-irradiation" are highlighted as primary innovation to distinguish this PIERS approach from "simultaneous illumination" in relevant references. This may be an important advance from practical consideration, however, still lack of enough new insight to fit with the outstanding standard required by high impact factor journal (Nature Communications).

Reviewer #2 (Remarks to the Author):

I think the changes in the manuscript now bring it to the point that it is suitable for publication.

Response to Reviewers

Reviewer #1 (Remarks to the Author):

The authors have reorganized the manuscript, and most of the technical issues are addressed with supplementary experiments. As a rebuttal by the authors, "pre-irradiation" are highlighted as primary innovation to distinguish this PIERS approach from "simultaneous illumination" in relevant references. This may be an important advance from practical consideration, however, still lack of enough new insight to fit with the outstanding standard required by high impact factor journal (Nature Communications).

We thank the reviewers for their comments on our manuscript. In our manuscript we show that this fundamental change in how the enhanced Raman spectra are collected, is instrumental in improved spectral collection and opens the door to a variety of new Raman substrates. We have demonstrated the new PIERS effect, discussed and presented evidence for the mechanism of the effect, including the evidence of a chemical enhancement, and have shown it to be highly effective against several practical detection challenges. We feel that this merits the work's inclusion in a high impact publication.

Reviewer #2 (Remarks to the Author):

I think the changes in the manuscript now bring it to the point that it is suitable for publication.

We thank the reviewer for their comments and time and are pleased they are satisfied with our account of the work.